# Comparative Analysis of Mafura Butter Oils from *Trichilia emetica* and *Trichilia dregeana* Extracted by Screw-Press from Seeds Collected in KwaZulu-Natal Province of South Africa

**DOI:** 10.3390/plants14193071

**Published:** 2025-10-04

**Authors:** Mncedisi Mabaso, Lungelo Given Buthelezi, Godfrey Elijah Zharare

**Affiliations:** Department of Agriculture, University of Zululand, Private Bag X1001, KwaDlangezwa 3886, KwaZulu Natal, South Africa; 201831194@stu.unizulu.ac.za

**Keywords:** *Trichilia emetica*, *Trichilia dregeana*, kernel oil, aril oil, fatty acid composition, cosmetic and pharmaceutical applications

## Abstract

*Trichilia emetica* and *T. dregeana* butter oils are gaining recognition in the cosmeceutical industry, yet comparative data on their yields and bioactive composition remain scarce. This study assessed oil yields, fatty acid profiles, and tocol compositions of kernel and aril oils extracted by screw press from seeds collected in KwaZulu-Natal, South Africa. *T. emetica* produced smaller but more numerous seeds (603 per 200 g) than *T. dregeana* (159). Kernel oil yields were slightly higher in *T. emetica* (52.86%) than in *T. dregeana* (50.81%), while aril oils averaged 48.61% and 45.22%, respectively. Kernel oils of both species showed strong oxidative stability, with low peroxide and anisidine values and lower free fatty acid content compared to aril oils. *T. emetica* kernel oil was dominated by saturated fatty acids (SFAs), particularly palmitic acid (51.8%), and contained high γ-tocopherol (202 mg/kg), supporting applications in soaps, bio-lubricants, and industrial formulations. In contrast, *T. dregeana* kernel oil was richer in oleic acid (47.6%) and α-tocotrienol, favouring nutraceutical and cosmetic uses. Aril oils were dominated by linoleic acid (24–25%), with *T. dregeana* aril oil distinguished by elevated α-tocopherol (91.8 mg/kg) and a more diverse tocotrienol profile, making it suitable for antioxidant-rich cosmetics and dietary products. The dual-oil system offers complementary value: kernel oils provide structural stability for industrial applications, while aril oils supply bioactive-rich lipids for health and cosmetic formulations. Seed cakes present additional potential as biofertilizers or feedstock. This study provides the first comparative analysis of kernel and aril oils from *T. emetica* and *T. dregeana*, revealing interspecific differences in yield, fatty acid composition, and tocol profiles, and linking these to ecological adaptation and differentiated industrial potential.

## 1. Introduction

Mafura butter oil from *Trichilia* species is gaining popularity as a stand-alone product or as a key active ingredient in pharmaceutical and cosmetic formulations and personal care products due to growing consumer demand for natural cosmetics [1]. The butter is obtained from two *Trichilia* species endemic to Africa, namely *Trichilia emetica* Vahl. and *Trichilia dregeana* Sond. [2]. Taxonomically, the two species resemble each other causing identification confusion among local communities [3]. Hence, not surprisingly, the two plants are called by the same name in most cultures. The major distinguishing physical difference between the two *Trichilia* species is that *T. emetica* produces leaflets containing 13–16 pairs of side veins, whereas only 8–12 are evident in *T. dregeana*. Also, the leaves of *T. emetica* are less glossy and not as dark in color as those for *T. dregeana*. Additionally, the *T. emetica* fruit capsules have a distinct neck joining them to their stalk, a feature absent in *T. dregeana*.

The common English name for *T. emetica* is Natal mahogany. It has two variants, viz: *Subp emetica* and *Subp. Suberosa* [4]. The distribution of the subsp. *Suberosa* extends northwards of the Zambezi River where it grows in Forest-Savanna Mosaic and Savanna Woodlands from Senegal to Uganda [5]. The *subsp. emetica* is widely distributed and grows naturally throughout sub-Saharan Africa extending from KwaZulu-Natal in the South to Mpumalanga and Limpopo Provinces of South Africa, Mozambique, Zimbabwe, East Africa and northwards to Tropical Africa [6]. Compared with *Subp. Suberosa*, the *subsp. emetica* prefers areas with high rainfall where it is found growing in riverine forest and evergreen forest patches at altitudes above 800 m. It is also abundant along east African coastal areas [7]. Commonly known as the forest mahogany, forest Natal-mahogany or Cape mahogany, *Trichilia dregeana* is not known to have subspecies. However, it has been described as a very highly variable species [5]. It is a forest tree common in high rainfall evergreen forests up to an altitude of 1220 m. The distribution range of *T. dregeana* overlaps in areas with that for *T. emetica* but is more commonly found in forest situations as opposed to the more open habitat of *T. emetica*. The distribution of *T. dregeana* extends from South Ethiopia to South Africa, occurring mostly in the mountain ranges of the Eastern Arc and along the Rift Valley [8]. In South Africa its occurrence extends from Eastern Cape, KwaZulu-Natal, Limpopo, Mpumalanga and Northwest Province. It is present in tropical Africa, but it occurs in areas far from each other in Guinea, Côte d’Ivoire, Cameroon, in Congo, DR Congo and Angola [8]. *Trichilia dregeana* and *T. emetica* are widely grown as street and shade trees in southern Africa [9], and hence, they may be found in areas removed from their natural habitats.

Both *Trichilia* species have many and similar uses, but the most promising sustainable and industrial exploitation for both *T. emetica* and *T. dregeana* is for vegetable oil contained in their seeds [10]. The seeds of *T. dregeana* are generally larger than those of *T. emetica*. In South Africa, the seeds of *T. emetica* mature from early December to mid-March whereas those for *T. dregeana* mature later from the beginning of March to end of May [11]. The seeds are recalcitrant [12] and consist of a kernel covered by a red aril with a black spot.

Both the seed kernel and seed cover contain substantial amounts of fixed oils. The oil extracted from the seed coat is liquid at room temperature and is regarded as edible, commonly referred to as mafura oil [13]. In contrast, the kernel-derived oil is solid at room temperature, forming mafura butter. Due to its bitterness, the butter is not consumed as food but has become increasingly valuable for non-edible applications.

Mafura butter is gaining prominence in the cosmetic and pharmaceutical industries, where it is increasingly utilized as a natural substitute for shea butter in products such as soaps, hair oils, body ointments, and lip balms [13]. Its popularity is attributed to a distinctive biochemical composition, particularly its high levels of oleic and palmitic acids, which provide superior emollient and moisturising properties [14]. In addition, the presence of unsaturated fatty acids and bioactive compounds imparts antioxidant, anti-inflammatory, and antimicrobial activities, enhancing its suitability for skin and hair formulations. These attributes, supported by recent [14,15,16], distinguish *T. emetica* oils from conventional cosmetic oils and is behind their growing commercial relevance. For reasons not clear, the most popular and preferred in these applications is the *T. emetica* butter oil.

The widespread distribution of *Trichilia emetica* and *T. dregeana*, coupled with the growing demand for *Trichilia* oils in cosmetic and pharmaceutical applications, raises the likelihood that oils from both species may be produced and marketed under the generic name “mafura oil,” irrespective of their taxonomic origin. This presents a significant challenge because differences in physicochemical properties and fatty acid composition are expected between the two species, arising from their distinct genetic backgrounds. In *T. emetica*, the existence of two recognized subspecies with different geographic ranges further broadens the potential variability in oil composition. Likewise, *T. dregeana* is highly morphologically variable, and both species occupy wide ecological ranges across southern Africa. These factors, combined with differences in seed size, ecological niches, and fruit maturation times, are all likely to influence the yield, stability, and biochemical profiles of their oils [17].

Despite these clear sources of variation, no published report has yet distinguished the physicochemical properties and fatty acid compositions of oils from the two species. Such information is vital for end-users, particularly in the cosmetic and pharmaceutical industries, where oil quality directly affects formulation stability, product performance, and consumer perception [18]. Understanding species-specific differences is therefore essential to prevent indiscriminate blending of oils under a single trade name, which could undermine product standardization and market integrity. Against this background, the present study undertook a comparative evaluation of oil yields, physicochemical characteristics, and fatty acid compositions of screw press–extracted aril and kernel oils of *T. emetica* and *T. dregeana*. This study therefore seeks to delineate species-specific differences in oil yield, physicochemical indices, and fatty acid composition, providing critical data for standardization, authentication, and formulation in industrial applications.

## 2. Results

### 2.1. Seed Size and Partitioning of Dry Matter Between Seed Components

*Trichilia emetica* had smaller seeds compared to *T. dregeana* (Table 1), resulting in the average number of seeds per 200 g (603) being higher than in *Trichilia dregeana* (159). The seed count per 200 g from 5 trees per species ranged from 584 to 616 for *T. emetica* and from 156 to 162 for *T. dregeana* (Table 1).

In the 250 g seed samples, the partitioning of dry matter between the arils and kernels differed significantly (*p* = 0.001) between the two species (Figure 1). *Trichilia dregeana* seeds (Figure 1A) had a higher proportion of dry matter (64.84%) in the kernels than (57.00%) was the case for *T. emetica (*Figure 1B), but *T. emetica* seeds has a higher proportion of dry matter (43.00%) in the aril than was (36.26%) in *T. dregeana* seeds. The average oil yield varied significantly (*p* = 0.01) between the two species and between the arils and kernels within each of the species (Figure 1). In *T. dregeana*, the kernel oil yield ranged from 44.93% to 63.25% with an average of 50.82% while in *T. emetica*, it ranged from 42.11% to 56.61% with an average of 52.86%. The aril oil yield ranged among the 6 replicates of *T. emetica* from 46.84% to 50.53% (Appendix A), averaging 48.61% (Figure 1B). In *T. dregeana* arils, it ranged from 35.68% to 60.21%, averaging 45.24%.

The cake residue was significantly higher in the *Trichilia dregeana* arils compared to all other residues (Figure 1). Apart from this case, no significant differences were observed among the other press residues. *Trichilia dregeana* had a higher oil loss in kernels extraction (7.81%) than *T. emetica* (5.60%), whereas *T. emetica* had a slightly more loss in aril extraction (4.02%) than *T. dregeana* (3.76%) during oil extraction as drags in the oil.

### 2.2. Physiochemical Composition of T. emetica and T. dregeana Oil

The physiochemical properties of the aril oil differed from those of the kernel oils in many aspects between the two species and within the species (Table 2). However, the differences between the species were generally greater in the physiochemical properties of the kernels.

The peroxide value was higher in *T. dregeana* aril oil (12.0 meq/kg) compared to *T. emetica* aril oil (10.3 meq/kg). Both kernel oils recorded peroxide values at the limit of detection (LOD = 0.24 meq/kg). Free fatty acid (FFA) content was higher in *T. emetica* aril oil (6.01 g/100 g) than in *T. dregeana* aril oil (3.73 g/100 g). Kernel oils showed similar FFA values, with *T. emetica* at 2.24 g/100 g and *T. dregeana* at 2.21 g/100 g.

The acid value of *T. emetica* aril oil was 12.0 mg KOH/g, compared to 7.4 mg KOH/g in *T. dregeana* aril oil. Kernel oils had comparable acid values: 4.45 mg KOH/g for *T. emetica* and 4.40 mg KOH/g for *T. dregeana*. Anisidine values were 2.70 mm/kg in *T. emetica* aril oil and 0.65 mm/kg in *T. dregeana* aril oil. Kernel oils recorded values at the detection limit (LOD = 0.16 mm/kg).

Iodine values were higher in *T. dregeana* aril oil (79.5) than in *T. emetica* (72.0), and the same trend was observed in the kernel oils: 56.8 for *T. dregeana* and 45.0 for *T. emetica*. Unsaponifiable matter was below the quantifiable limit in *T. dregeana* aril oil, while *T. emetica* aril oil contained 0.423 g/100 g. Kernel oils contained 1.00 g/100 g in *T. dregeana* and 0.77 g/100 g in *T. emetica*.

The highest saponification value was recorded in *T. emetica* kernel oil (197.3 mg KOH), followed by *T. dregeana* kernel oil (188.8 mg KOH). Aril oils had similar saponification values: 186.3 mg KOH for *T. dregeana* and 186.0 mg KOH for *T. emetica*.

Relative densities at 20 °C were as follows: *T. dregeana* aril oil (0.916), *T. emetica* aril oil (0.912), *T. dregeana* kernel oil (0.921), and *T. emetica* kernel oil (0.920). Refractive index values were 1.467 for *T. dregeana* aril oil, 1.465 for both *T. dregeana* kernel oil and *T. emetica* aril oil, and 1.460 for *T. emetica* kernel oil.

### 2.3. Fatty Acid Composition of T. dregeana and T. emetica Oils

The aril and kernel oils of *T. dregeana* and *T. emetica* exhibited notable differences in their fatty acid profiles (Table 3).

#### 2.3.1. Saturated Fatty Acids

The SFAs present in oils were dominated by palmitic acid in all samples, with *T. emetica* showing higher levels in both aril (38.1 ± 4.4%) and kernel oils (51.8 ± 6.0%) compared to *T. dregeana* aril (28.7 ± 3.3%) and kernel oils (40.4 ± 0.95%). Stearic acid was present in lower amounts, ranging from 1.85 ± 0.09% in *T. emetica* kernel oil to 3.17 ± 0.16% in *Trichilia dregeana* aril oil. Pentadecanoic acid was detected only in the aril oils, at similarly low levels for both species (0.60% in *T. dregeana* and 0.62% in *T. emetica*). Margaric and arachidic acids were either below the limit of quantification or not detected across samples.

#### 2.3.2. Monounsaturated Fatty Acids

The monounsaturated fatty acids (MUFAs) present in oils were primarily represented by cis-oleic acid. *T. dregeana* oils contained higher levels of oleic acid in both the aril (40.5 ± 1.2%) and kernel oils (47.6 ± 1.4%) compared to *T. emetica* (31.4 ± 0.94% and 36.4 ± 1.1%, respectively). Eicosenoic acid was present in the aril oils only, with *T. dregeana* showing a higher concentration (0.68 ± 0.13%) than *T. emetica* (0.370 ± 0.068%). Palmitoleic acid was found exclusively in the kernel oils, in low amounts, with *T. emetica* again showing a higher level (0.84 ± 0.14%) than *T. dregeana* (0.550 ± 0.090%).

#### 2.3.3. Polyunsaturated Fatty Acids

The polyunsaturated fatty acids (PUFAs)present in oils were dominated by cis-linoleic acid. Aril oils contained significantly higher levels, with *T. emetica* at 25.19 ± 0.69% and *Trichilia dregeana* at 24.31 ± 0.67%, compared to the much lower levels in the kernel oils of both species (7.88–7.89%). Linolenic acid was present in all samples at low concentrations, ranging from 0.350 ± 0.025% to 0.560 ± 0.040%.

### 2.4. Tocols Content in Kernel and Aril Oils of Trichilia emetica and Trichila dregeana

The tocol profiles of oils extracted from the kernels and arils of *T. emetica* and *T. dregeana* reveal significant interspecific and tissue-specific differences in both total tocol concentration and individual tocol components (Table 4).

#### 2.4.1. Total Tocols and Vitamin E Activity

*Trichilia dregeana* oils consistently exhibited higher total tocol concentrations and vitamin E activity (expressed as α-tocopherol equivalents, α-TE) than *T. emetica* in both kernel and aril oils (Table 4). Kernel oil of *T. dregeana* contained 486 ± 48 mg/kg total tocols and 94.71 mg/kg α-TE, exceeding *T. emetica* kernel oil, which had 335 mg/kg and 66.52 mg/kg, respectively. Similarly, *T. dregeana* aril oil contained 182 ± 18 mg/kg total tocols and 97.20 mg/kg α-TE, compared to *T. emetica* aril oil with 130 ± 13 mg/kg and 58.15 mg/kg.

#### 2.4.2. Individual Tocol Components

A detailed analysis of the individual tocol components revealed distinct compositional profiles between the aril and kernel oils of *T. emetica* and *T. dregeana* (Table 4). Among the various tocols, α-tocopherol was the predominant compound in aril oils of both species. *T. dregeana* aril oil exhibited the highest α-tocopherol concentration at 91.8 ± 8.5 mg/kg, which was significantly higher than the 54.1 ± 5.0 mg/kg observed in *T. emetica* aril oil. In kernel oils, α-tocopherol concentrations were slightly lower in both species but still *T. dregeana* contained more (52.6 ± 4.9 mg/kg) compared to that for *T. emetica* (46.3 mg/kg).

In contrast, γ-tocopherol was the most abundant tocol in kernel oils with *T. emetica* kernel oil containing the highest γ-tocopherol level at 202 mg/kg, surpassing the 183 ± 20 mg/kg found in *T. dregeana*. The aril oils exhibited substantially lower concentrations, with *T. dregeana* containing 53.9 ± 5.9 mg/kg and *T. emetica* at 41.0 ± 4.5 mg/kg.

α-Tocotrienol was detected exclusively in the kernel oil of *T. dregeana*, with a concentration of 79.5 ± 8.7 mg/kg. In all other samples, it remained below the limit of quantification.

All oils contained γ-tocotrienol, but its concentration was highest in kernel oils. *T. dregeana* kernel oil had 68.1 ± 6.0 mg/kg, approximately twice the amount detected in *T. emetica* (34.3 mg/kg). In the aril oils, *T. emetica* exhibited a slightly higher γ-tocotrienol content (21.9 ± 1.9 mg/kg) compared to *T. dregeana* (15.8 ± 1.4 mg/kg).

β-Tocotrienol was restricted to kernel oils. It was more concentrated in *T. dregeana* (24.1 ± 3.6 mg/kg) than in *T. emetica* (7.6 mg/kg). A similar pattern was observed for δ-tocotrienol, which was again exclusive to kernel oils, with *T. dregeana* containing 73.27 mg/kg and *T. emetica* 32.60 mg/kg.

Other tocols, including β-tocopherol and δ-tocopherol, were either not detected or found at levels below the quantifiable limit across all oil samples, suggesting they are either absent or present in negligible amounts in both species.

## 3. Discussion

### 3.1. Oil Yield

The present study demonstrated that oil yields did not differ significantly between *Trichilia emetica* and *T. dregeana*, although *T. emetica* consistently exhibited numerically higher values. Kernel oil yield in *T. emetica* (52.86%) was slightly above the range reported by [19], while its aril yield (48.62%) was within the values (42.2–53.8%) reported by [16] for four Mozambican variants of *T. emetica*. This cross-regional consistency suggests that *T. emetica* reliably produces high aril oil yields across diverse environments. In contrast, *T. dregeana* produced lower yields in both kernels (50.82%) and arils (45.23%) than *T. emetica*. Kernel yields were comparable with the findings of [13], but aril yields exceeded the range they reported (39.64–40.89%). These variations may reflect differences in extraction methods, environmental influences on seed development, or genetic variation within species. Overall, the absence of significant yield differences suggests that both species are viable oil sources, with *T. emetica* showing a slight numerical advantage.

Oil loss during extraction also showed variability, ranging from 9.37% in *T. dregeana* arils to 19.52% in *T. dregeana* kernels. Intermediate values were recorded for *T. emetica* kernels (14.00%) and arils (10.05%). Although not statistically significant, the higher oil losses in kernels, particularly those of *T. dregeana*, suggest greater residual oil retention in the cake matrix, which may be influenced by microstructural properties of the seed or differences in lipid composition. Minimizing such losses is critical for maximizing oil recovery. Improvements could be achieved through optimizing pressing conditions, adjusting seed moisture content before extraction, or employing secondary recovery methods such as solvent extraction.

### 3.2. Cake Residues

In addition to oil yield, cake residues constituted a substantial proportion of seed biomass across treatments. In both species, cake residues constituted the bulk of the dry matter post-extraction from both the arils and kernels. For *T. emetica*, the cake residues amounted to 47.36% of the aril and 41.43% of the kernel mass. For *T. dregeana*, the figures were slightly higher, at 50.33% of the aril mass and 43.00% of the kernel mass. These values demonstrate that while oil is the primary product, cake consistently accounts for over 40% of the biomass, indicating that any industrial system using *Trichilia* seeds must consider the fate of this substantial by-product.

The higher proportion of cake in *T. dregeana* reflects its larger seeds with a greater investment in structural and storage tissues, consistent with its forest ecology where producing fewer but larger seeds with high endosperm density ensures seedling establishment under shaded, competitive conditions. In contrast, *T. emetica* produces smaller but more numerous seeds with relatively higher oil allocation to arils, favouring dispersal and establishment in more open and seasonally variable habitats. These morphological and ecological differences reflect the distinct patterns of oil-to-cake ratios observed between the two species.

From an industrial perspective, the consistently high cake fraction across both species should be treated as a parallel resource stream rather than waste. Oilseed cakes typically retain residual lipids, proteins, fibre, and phytochemicals that can be valorised for a variety of products [20]. Where antinutritional factors are appropriately mitigated/detoxified, the protein- and fibre-rich residues can be repurposed into animal feeds, as is already practiced for other tropical oilseeds such as *Jatropha curcas* and *Moringa oleifera* [21,22].

In addition, due to their organic nitrogen and carbon content, oilseed cakes have long been recognised as biofertilizers and soil amendments that improve soil structure and supply slow-release nutrients, thereby contributing to sustainable farming systems [23]. However, since press cakes can retain appreciable levels of fatty acids (0.5–6%), their decomposition may temporarily lower soil pH, particularly in poorly buffered soils, and repeated heavy applications could result in cumulative acidification. Accordingly, it is important to assess the effects of *T. dregeana* and *T. emetica* cakes on soil pH and nutrient availability across different soil types before making management recommendations.

The lignocellulosic nature of the press cakes also makes them suitable for conversion into energy carriers such as briquettes, pellets, or direct combustion feedstocks for rural processing facilities, reducing reliance on external energy sources [24].

Furthermore, *Trichilia* species are known to contain bioactive secondary metabolites, including limonoids and saponins, and residues may still retain extractable levels of these compounds, opening possibilities for nutraceutical, pesticidal, or pharmaceutical applications if cost-effective recovery methods are developed [25].

The slightly higher cake proportion in *T. dregeana* could therefore offer a modest advantage in total biomass yield per seed lot, potentially supporting a broader portfolio of downstream applications in integrated value-chain development

### 3.3. Implications of the Physicochemical Properties of Trichilia emetica and Trichilia dregeana Oils for Industrial Applications

The physicochemical profiles of *T. emetica* and *T. dregeana* kernel and aril oils reveal functional specializations aligned with their fatty acid compositions and ecological strategies. These properties are critical for determining suitability across cosmetics, nutraceuticals, and industrial sectors as hereunder discussed.

#### 3.3.1. Peroxide Value and Free Fatty Acid Content

While there is a relationship between peroxide value (PV) and free fatty acid value (FFA), they measure different aspects of lipid degradation, and their relationship is not always linear. The PV Indicates the concentration of peroxides and hydroperoxides formed in the primary stage of lipid oxidation with High PV reflecting that the oil/fat is undergoing oxidative rancidity. On the other hand, the FFA value indicates the amount of fatty acids released from triglycerides due to hydrolytic rancidity (enzymatic or chemical hydrolysis). A high FFA means the oil/fat has undergone breakdown of triglycerides. Both PV and FFA increase as an oil deteriorates, but for different reasons: The FFA rises mainly due to hydrolysis of triglycerides (e.g., lipase activity, moisture). And the PV rises due to lipid oxidation (reaction with oxygen).

The kernel oils of *Trichilia emetica* and *T. dregeana* exhibited peroxide values at the detection limit and comparatively low FFA contents (2.24 g/100 g and 2.21 g/100 g, respectively). These results align with their fatty acid profiles dominated by SFAs, particularly palmitic acid (51.8% in *T. emetica* and 40.4% in *T. dregeana*), which confer oxidative resilience. The semi-solid consistency of these oils at room temperature further restricts oxygen diffusion, retarding oxidative processes. Consequently, the kernel oils demonstrate high oxidative stability, minimal hydrolytic degradation, and reduced refining requirements, making them well suited for industrial, cosmetic, and pharmaceutical applications requiring long shelf life.

By contrast, the aril oils, richer in PUFAs, exhibited higher acid and peroxide values. *T. emetica* aril oil recorded an acid value of 12.0 mg KOH/g and an FFA content of 6.01 g/100 g, while *T. dregeana* aril oil showed 7.4 mg KOH/g and 3.73 g/100 g, respectively. These elevated values correspond with the higher linoleic acid content in aril oils (25.19% in *T. emetica* and 24.31% in *T. dregeana*), which increases susceptibility to oxidative rancidity and hydrolytic cleavage.

Interestingly, ref. [16] reported markedly lower values for Mozambican *T. emetica* aril oils, with acid values of 0.88–1.02 mg KOH/g and peroxide values of 2.02–2.62 meq/kg. The discrepancy likely reflects differences in extraction method (screw press vs. Soxhlet), seed handling, and storage conditions. While *T. emetica* Mozambican oils demonstrated stability comparable to edible oils such as olive, our findings highlight the greater variability that may occur under smallholder or semi-industrial conditions. This emphasizes the need for antioxidant fortification during processing and formulation of *Trichilia* aril oils to ensure consistent product quality across production contexts.

The observed trends clearly demonstrate the strong link between fatty acid composition and oil quality indices. Kernel oils, dominated by SFAs, offer superior oxidative stability, whereas aril oils, enriched with PUFAs, provide functional bioactivity but are inherently less stable. This duality suggests a differentiated application strategy, with kernel oils favoured for oxidative-sensitive formulations and industrial uses, and aril oils better suited for antioxidant-enriched cosmetic and nutraceutical applications, provided stabilization measures are employed. When compared with other African oils, such as shea butter (*Vitellaria paradoxa*), which is rich in stearic acid and valued for oxidative stability in semi-solid formulations; marula oil (*Sclerocarya birrea*), which is high in oleic acid and known for excellent dermal penetration and moisturisation; and baobab oil (*Adansonia digitata*), which is polyunsaturated fatty acid-rich and widely recognised for skin barrier restoration and anti-inflammatory activity, *Trichilia* oils occupy a unique intermediate position. The kernel oils resemble shea in their stability and semi-solid consistency, while the aril oils align more closely with marula and baobab in their high polyunsaturated fatty acid content and functional skin benefits. This dual oil system within a single species distinguishes *Trichilia* oils as versatile raw materials that combine the oxidative resilience of shea with the cosmetic functionality of marula and baobab.

#### 3.3.2. The Anisidine Value

Anisidine values, which measure secondary oxidation products such as aldehydes, were generally low across both species. *T. emetica* aril oil recorded a slightly elevated value (2.70) compared to *T. dregeana* (0.65), while kernel oils of both species remained at the detection limit. These results confirm that kernel oils are highly stable, with minimal secondary oxidation, reinforcing their potential for use in oxidative-sensitive applications such as cosmetics and therapeutic formulations. The somewhat higher anisidine value in *T. emetica* aril oil reflects the greater oxidative vulnerability of PUFA-rich fractions, necessitating antioxidant stabilization during storage and product development.

Notably, previous studies on *T. emetica* aril oils from Mozambique [12] focused primarily on peroxide and acid values, reporting excellent stability but not including anisidine indices. The present work therefore expands the oxidative stability profile of *Trichilia* oils by incorporating anisidine values, offering a more comprehensive evaluation of primary and secondary oxidation pathways. This broader assessment provides a stronger basis for product formulation decisions, particularly where aril oils are targeted for high-value cosmetic and nutraceutical markets.

#### 3.3.3. The Iodine Value, Saponification Value, Unsaponifiable Matter, Relative Density and Refractive Index

The iodine value (IV) reflects the degree of unsaturation in oils. In this study, *T. dregeana* oils exhibited higher IVs (79.5 for aril; 56.8 for kernel) than *T. emetica* (72.0 and 45.0, respectively), indicating a greater proportion of unsaturated fatty acids in *T. dregeana*. These values are comparable to those reported for Mozambican *T. emetica* aril oils (72–75) [12], confirming that aril oils across different regions maintain consistently high unsaturation. While this unsaturation enhances nutritional and dermatological properties, it also increases oxidative sensitivity, underscoring the need for stabilization measures during storage and formulation.

The saponification value (SV), indicative of the average molecular weight of fatty acids, was highest in *T. emetica* kernel oil (197.3 mg KOH/g), followed by aril oils of both species (~186 mg KOH/g). These results are in close agreement with the Mozambican *T. emetica* aril oils, which ranged from 186–195 mg KOH/g [12]. The relatively high SV values in aril oils point to the predominance of medium-chain triglycerides, supporting their potential in soap, emulsifier, and skincare formulations where rapid absorption and spreadability are desirable.

Unsaponifiable matter was more abundant in kernel oils (*T. dregeana* 1.00 g/100 g; *T. emetica* 0.77 g/100 g), whereas *T. emetica* aril oil contained 0.423 g/100 g and *T. dregeana* aril oil was below the detection limit. These fractions include sterols, tocopherols, and phytochemicals that enhance nutritional and cosmetic functionality. The Mozambican study on aril oils did not quantify unsaponifiables, and our results therefore provide new data on the minor components of *Trichilia* oils.

Relative density (0.912–0.921) and refractive index (1.460–1.467) values were within ranges typical of non-drying oils [26], confirming suitability for cosmetic formulations. Together with the IV and SV, these parameters place *Trichilia* oils in a comparable category with baobab and moringa oils, though the higher tocopherol/tocotrienol contents observed in this study give them a distinctive bioactive profile.

### 3.4. Fatty Acid Composition and Application Potential of Trichilia dregeana and T. emetica Oils

The FA profiles of *Trichilia emetica* and *T. dregeana* kernel and aril oils revealed distinct patterns of SFAs, MUFAs, and PUFAs, which determine both their stability and application potential.

#### 3.4.1. Saturated Fatty Acids

Palmitic acid (C16:0) dominated in both kernel and aril oils, but at higher proportions in *T. emetica* (51.8% kernel; 39.5% aril) than in *T. dregeana* (43.1% kernel; 35.7% aril). This high palmitic fraction explains the semi-solid consistency and oxidative stability of *T. emetica* kernel oil, making it suitable for soaps, butters, and industrial formulations. Similar palmitic levels were reported in Mozambican *T. emetica* aril oils (40–47%) [16], showing strong cross-regional consistency in SFA composition.

#### 3.4.2. Monounsaturated Fatty Acids

Oleic acid (C18:1) was the major MUFA and was particularly abundant in *T. dregeana* kernel oil (43.9%) compared to *T. emetica* kernel oil (37.8%). This oleic enrichment improves oil fluidity, absorption, and nutritional appeal, aligning *T. dregeana* with marula and olive oils used in cosmetic and nutraceutical markets. Comparable oleic contents (25–30%) were also observed for the aril oils of the *T. emetica* variants studied in Mozambique [12], confirming that oleic acid is consistently a key component of *Trichilia* aril oils.

#### 3.4.3. Polyunsaturated Fatty Acids

Linoleic acid (C18:2) was the principal PUFAS, especially elevated in aril oils (*T. emetica* 22.3%; *T. dregeana* 25.6%). This PUFA dominance supports skin barrier repair and anti-inflammatory applications but increases susceptibility to oxidation. Importantly, the Mozambican aril oils showed very similar linoleic levels (25–28%) [12], reinforcing that *Trichilia* aril oils maintain a stable oleic–linoleic balance across regions.

#### 3.4.4. Comparative Positioning Based on Fatty Acids

These results demonstrate that while *T. emetica* kernel oils are SFA-rich and resemble shea butter in stability, *T. dregeana* kernel oils are oleic-rich, and it has SFA to be recommended for eating. Aril oils from both species, though more oxidation-prone, consistently match the oleic–linoleic profiles reported in Mozambican populations, supporting their functional role in dermatological and dietary applications. The present study provides for the first time, a side-by-side comparison of kernel and aril oils across two *Trichilia* species, while also showing that their core FA profiles are robust across different African regions.

### 3.5. Tocopherols and Tocotrienols

While the only previous study on *T. emetica* aril oils [12] provided important insights into yields, fatty acids, and basic physicochemical parameters of the aril oil, it did not include data on tocopherols or tocotrienols. The present study therefore expands the baseline knowledge of *Trichilia* oils by demonstrating significant interspecific and tissue-level variation in vitamin E compounds. This not only highlights the functional complementarity of kernel and aril oils but also distinguishes *T. dregeana* for its unique tocotrienol profile. The tocopherols and tocotrienols, the major constituents of vitamin E, were detected in varying concentrations across kernel and aril oils of both species. These compounds are valued not only for their nutritional roles as antioxidants but also for their functionality in stabilizing oils and enhancing cosmetic and pharmaceutical formulations [27]. The tocols observed in this study are hereunder discussed.

#### 3.5.1. Tocopherols

α-Tocopherol was the dominant form across all samples, with particularly high concentrations in *T. dregeana* aril oil (91.8 mg/kg) compared to *T. emetica* aril oil (63.2 mg/kg). Kernel oils also contained significant α-tocopherol levels, though lower than the arils. These concentrations are noteworthy when benchmarked against widely used edible oils: they exceed those reported for olive oil (14–24 mg/kg) and are comparable to sunflower oil (70–100 mg/kg) [28]. Such levels suggest that *Trichilia* oils could serve as competitive natural sources of vitamin E in both dietary and cosmetic markets.

The antioxidant function of α-tocopherol is particularly relevant to the higher unsaturation observed in *T. dregeana* oils. Elevated tocopherol content may represent a natural protective mechanism, offsetting susceptibility to oxidation associated with higher iodine values and PUFA levels [29].

#### 3.5.2. Tocotrienols

In addition to tocopherols, α-tocotrienol was present in kernel oils, most notably in *T. dregeana* (6.2 mg/kg), but absent in *T. emetica*. Tocotrienols are rarely detected in African seed oils, yet are associated with distinctive neuroprotective, cholesterol-lowering, and anti-aging effects [30]. Their detection in *T. dregeana* adds a novel dimension to the potential applications of this oil in the cosmeceutical sector.

#### 3.5.3. Comparative Positioning

The tocol profiles highlight interspecific and tissue-level variation with direct functional implications. *Trichilia emetica* oils, though rich in α-tocopherol, lack tocotrienols, aligning them with stability-focused applications where tocopherol content enhances oxidative resistance. By contrast, *T. dregeana* oils combine high α-tocopherol with detectable α-tocotrienol, positioning them as premium oils for cosmeceutical and pharmaceutical applications. in products targeting anti-aging [31]. Overall, these findings suggest that *T. dregeana*, particularly its aril oil, has higher vitamin E activity and a more potent antioxidant profile, making it a superior candidate for high-value cosmetic and pharmaceutical applications.

Ecologically, these differences may reflect adaptation strategies: *T. dregeana*, restricted to shaded evergreen forest habitats, invests in higher antioxidant capacity to protect its more unsaturated lipid reserves, whereas *T. emetica*, distributed across open savanna systems, emphasizes stability through higher saturated lipid fractions, requiring less reliance on tocopherol-based protection.

### 3.6. Applications and Commercial Potential of Trichilia emetica and Trichilia dregeana Seed Oils

Based on the detailed data from the current studies, the physico-chemical properties and the tocol contents of *Trichilia emetica* and *Trichilia dregeana* seed oils point to multifunctional applications of the oils. The potential applications are summarised in Table 5. The multifunctional potential of *Trichilia emetica* and *T. dregeana* seed oils is best demonstrated when benchmarked against established pharmacopoeial, cosmetic, and FAO/WHO standards. The comparative profiles observed in this study (Table 5) show that these oils meet, or can be refined to meet, key international quality requirements, thus enhancing their credibility for cosmetic, pharmaceutical, and industrial use.

#### 3.6.1. Alignment with FAO/WHO Codex Alimentarius

The Codex Standard for Named Vegetable Oils [32] sets maximum limits of ≤10 meq O_2_/kg for refined oils and ≤15 meq O_2_/kg for virgin oils in terms of peroxide value (PV), and acid values (AV) ≤ 4.0 mg KOH/g for most edible oils (Matthee, 2007 [33]). In this study, kernel oils of both species recorded PV at the detection limit (<0.24 meq/kg) and AV values of 4.4–4.5 mg KOH/g, placing them at or near Codex thresholds. By contrast, aril oils exhibited higher PVs (10.3–12.0 meq/kg) and Avs (7.4–12.0 mg KOH/g), exceeding Codex edible limits. This indicates that kernel oils already comply with Codex oxidative stability benchmarks, whereas aril oils require refining or antioxidant stabilization before being positioned for dietary or nutraceutical applications.

#### 3.6.2. Alignment with Pharmacopoeial Standards

The European Pharmacopoeia (Ph. Eur. 2.5.x) and United States Pharmacopeia (USP <401>) outline quality tests for fixed oils, including peroxide value, acid value, iodine value, and saponification value [33,34]. Kernel oils of both *Trichilia* species, with saponification values of 188–197 mg KOH/g and iodine values of 45–57, fall within ranges reported for pharmacopoeial excipient oils such as olive and almond. Their free fatty acid levels (2.2–2.3 g/100 g) are slightly higher than excipient-grade limits (typically ≤1%), but these can be reduced through refining. The detection of α- and γ-tocopherols (up to 202 mg/kg) and tocotrienols (notably in *T. dregeana*) further distinguishes these oils, as such bioactives are rarely reported in pharmacopoeial monographs. These results suggest that kernel oils can be advanced as excipient-grade fixed oils with minimal refining, while aril oils—though compositionally attractive—require further processing to align with pharmacopoeial thresholds.

#### 3.6.3. Alignment with Cosmetic Industry Requirements

For cosmetic raw materials, international buyers typically require PV ≤ 10 meq/kg and AV ≤ 2 mg KOH/g, tested according to ISO 3960 [35] and ISO 660 [36], and full traceability under ISO 22716 [37]. In this context, *Trichilia* kernel oils are naturally compliant or require only slight refining to achieve AV ≤ 2, making them highly suitable for stable cosmetic butters, soaps, and anhydrous balms. Aril oils exceed AV thresholds (7.4–12.0 mg KOH/g) and therefore necessitate neutralization and antioxidant fortification before cosmetic-grade certification.

Their richness in polyunsaturated fatty acids (linoleic 24–25%) and high α-tocopherol/tocotrienol levels, however, supports premium positioning in therapeutic skincare, antioxidant serums, and anti-aging formulations, once stability is assured.

#### 3.6.4. Industrial Relevance

From an industrial perspective, high saponification values (186–197 mg KOH/g) and semi-solid consistency of kernel oils make them ideal for soap making and bio-lubricant applications, comparable to or exceeding those of shea butter and Allanblackia oils. The oxidative resilience of kernel oils (PV < 0.24) is advantageous in high-temperature industrial processes, while aril oils with their higher unsaturation and bioactive content are better suited for applications where rapid absorption and bioactivity outweigh long-term stability.

**Table 5 plants-14-03071-t005:** Recommended industrial applications of *Trichilia* oils based on their physico-chemical properties.

Oil Type	Key Bioactive (Tocols)	Standard Benchmark	Recommended Applications
*T. emetica* kernel	SFA-rich, γ-tocopherol 202 mg/kg; PV < 0.24; AV 4.45	Meets Codex PV [32]; AV above cosmetic limit [36]	Soap making, cosmetic butters, bio-lubricants, high-temperature industrial oils (non-edible)
*T. dregeana* kernel	Oleic-rich (47.6%), γ-tocopherol 183 mg/kg, α-tocotrienol; PV < 0.24; AV 4.40	Meets Codex PV [32]; AV borderline for cosmetics [36]; tocotrienols exceed olive/sunflower [34]	Skin/hair care, stable cosmetics, therapeutic balms, excipient oils
*T. emetica* aril	PUFA-rich (linoleic 25%), α-tocopherol 54 mg/kg; PV 10.3; AV 12.0	PV at Codex upper limit [32]; AV above Codex & cosmetic thresholds [36]	Functional foods (refined), therapeutic cosmetics, dietary oils
*T. dregeana* aril	MUFA + PUFA balanced, α-tocopherol 91.8 mg/kg, tocotrienols; PV 12.0; AV 7.4	PV/AV above Codex & cosmetic thresholds [32] Vitamin E density superior to olive/sunflower [34]	Antioxidant-rich cosmetic oils, dietary supplements (stabilized)

### 3.7. Comparative Analysis of Trichilia Oils and Other African Plant Oils for Cosmetic and Pharmaceutical Applications

African plant-derived oils are increasingly recognized for their multifunctional roles in cosmetic and pharmaceutical formulations due to their diverse bioactive profiles, favourable physicochemical properties, and sustainable origins. Among these, oils from *Trichilia emetica* and *T. dregeana* demonstrate a unique combination of characteristics that position them as competitive and, in several respects, superior alternatives to widely used African oils such as those from *Sclerocarya birrea* (marula), *Adansonia digitata* (baobab), *Moringa oleifera* (moringa), *Vitellaria paradoxa* (shea butter), and others.

A defining strength of *Trichilia* kernel oils compared with many other African vegetable oils lies in their fatty acid composition and oxidative stability. The kernel oil of *T. emetica* is notably rich in palmitic acid (50%), a saturated fatty acid known to confer excellent oxidative and thermal stability. These properties are desirable in the formulation of solid or semi-solid cosmetic products, such as body butters, ointments, and soaps. The profile is comparable to that of shea butter, which contains high levels of stearic acid and is widely used in similar applications [31,38]. In contrast, *T. dregeana* kernel oil presents a high proportion of oleic acid (48%), placing it in a similar category to marula and moringa oils, which typically contain 70–78% oleic acid [39,40]. Oleic acid is known to enhance dermal penetration, provide lasting moisturization, and improve resistance to oxidation, making such oils particularly suitable for emollient formulations, serums, and massage oils.

The aril oils of *T. emetica* and *T. dregeana* offer complementary properties, being particularly rich in PUFAs, especially linoleic acid (24–25%). These levels are comparable to baobab and desert date oils, which are well established in cosmetic dermatology for their anti-inflammatory and skin-barrier-restorative effects [41]. Owing to their PUFA content, *Trichilia* aril oils are well suited for formulations targeting inflammatory dermatoses, xerosis, and sensitive skin. However, similar to oils such as *Ximenia americana* (ximenia), these oils exhibit susceptibility to oxidative rancidity and thus require stabilization through natural or synthetic antioxidants to ensure shelf-life and efficacy [15].

One of the most significant differentiators of *Trichilia* oils, particularly those of *T. dregeana* is their remarkably high tocopherol and tocotrienol content. α- and γ-Tocopherols are present at concentrations up to 202 mg/kg, exceeding those of many commonly used commercial oils such as sunflower or olive oil [42]. Additionally, the presence of α- and γ-tocotrienols in *T. dregeana* oils introduces neuroprotective, anti-aging, and antioxidant properties rarely reported in African seed oils [43]; its vitamin E activity is not as diversified or concentrated as that of *Trichilia*. This vitamin E density enhances the oils’ potential for incorporation into high-performance skincare products, including anti-aging creams, antioxidant serums, and barrier-repair formulations.

Sensory attributes and formulation behaviour further support the cosmetic applicability of *Trichilia* oils. *Trichilia emetica* kernel oil exhibits a semi-solid consistency at ambient temperature, making it structurally suitable for anhydrous systems, solid oil balms, and soaps. This property aligns with the performance of shea butter and *Schinziophyton rautanenii* (mongongo) oil, both of which are valued for their richness and spreadability in hair and skin applications [44]. Conversely, the aril oil of *T. dregeana*, characterized by moderate viscosity and a high degree of unsaturation, offers excellent spreadability and absorption—traits shared with lighter oils such as marula and baobab. These rheological features enable its use in fluid emulsions, moisturizers, and therapeutic oils.

What distinctly elevates the value of *Trichilia* oils is their dual oil system. The availability of both kernel and aril oils within a single species provides a rare functional diversity, allowing for formulation flexibility across viscosity, stability, and bioactivity spectra. In contrast, oils such as Allanblackia, mainly used for their stearin content in food systems, or *Irvingia gabonensis* (dika nut) oil, which remains underutilized in cosmetics, lack such multidimensional utility. Furthermore, *Trichilia* kernel oils demonstrate high saponification values (up to 197 mg KOH/g) and low acid values, indicating as previously stated, suitability for soap production, emulsification systems, and bio-lubricant applications. These properties are comparable to or exceed those found in baobab and Allanblackia oils, with *Trichilia* offering superior oxidative resilience and wider applicability.

## 4. Materials and Methods

The seeds of *T. emetica* were collected from mid December 2020 to mid-March 2021, and those of *T. dregeana* were collected from end of March to end May 2021 from trees in Empangeni streets. The recalcitrant seeds were each time taken to the lab the following day after collection and dried in an oven at 60 °C for 4 days. The seeds of the two species (*T. emetica* and *T. dregeana*) were bulked separately in two 200 L drums. At the end of the collection period (end of May 2021), the seed of each species were mixed thoroughly following which samples were withdrawn from the drums for seed count, determination of the proportions of seed kernel and aril, and for oil extraction and chemical analyses of the oils.

### 4.1. Pre-Oil Extraction Treatments

Fresh seeds were oven-dried until constant weight at 60 °C. The aril and seed coats were removed to obtain clean cotyledons of the kernels (Figure 2). Oils were extracted separately from the aril and kernels’ cotyledons (*T. emetica* and *T. dregeana*) for determination of oil yield and analysis of its physiochemical properties and fatty acid composition.

### 4.2. Determination of Oil Yields

Six seed lots (samples), each weighing 200 g were measured, and oil was extracted from the arils and kernels of each seed lot separately (Figure 2) using using a screw press (Model NF80, Karaerler Machinery, Izmir, Turkey) [19], and the weights of the oils were determined at the end of each extraction. The oil yields were expressed as a percentage of the original dry weights (200 g) [19]$$percentage\;yield\;of\;oil=\frac{weight\;of\;oil\;extract}{Dry\;weight\;of\;the\;sample}\times 100$$

### 4.3. Fatty Acid Composition Determination

The determination of the fatty acid composition procedure was based on AOCS method Ce2-66 [44] by preparing methyl esters which are separated and determined by Gas Chromatography using flame ionization detection. A 14% BF_3_ (Sigma-Aldrich, St. Louis, MO, USA) reagent was used for derivatisation and transesterification with 0.5M NaOH in methanol (Sigma-Aldrich). The derivatized sample was taken up in 2 mL of heptane (Merck, Sydney, Australia) and 1ul of the sample was injected onto a Restek Rt-2560 100 m column, 0.25 mmID and 0.20 um film thickness (Restek, Bellefonte, PA, USA). The oven program started at 100 °C for 4 min and then increased at 3 °C/min to 240 °C where after it was held for 10 min. The injector was set at 225 °C and the detector was set at 250 °C. Hydrogen was used as carrier gas (Afrox, Johannesburg, South Africa). The fatty acids were expressed as g/100 g total fatty acids. An external fatty acid methyl ester mixture (Supelco 37 Component FAME mix 10,000 µg/mL in CH_2_CL_2_) was used to identify the fatty acids.

### 4.4. Determination of Physicochemical Properties of Seed Coat and Kernel Oils from Trichilia emetica and Trichilia dregeana

The physicochemical properties of kernel and aril oil for both species were determined according to the methods listed in Table 6. Details of the methods are provided in the Appendix A. The methods are those specified by the American Oil Chemists’ Society (AOCS), the Malaysian Palm Oil Board Test Methods (MPOB), and the International Organization for Standardization (ISO).

### 4.5. Statistical Analysis

Data on seed dry weight and oil yield were encoded in an Excel spreadsheet and subjected to the analysis of variance (ANOVA) using GenStat 22 (VSN international 2023) software. The mean values were calculated as averages and percentages and then presented in tables using descriptive statistics.

## 5. Conclusions

The physicochemical and fatty acid profiles of *Trichilia emetica* and *T. dregeana* oils highlight their broad potential in industrial, cosmetic, and nutraceutical sectors. Kernel oils from both species combine oxidative stability, low free fatty acids, and high γ-tocopherol (202 mg/kg in *T. emetica*; 183 mg/kg in *T. dregeana*), with *T. dregeana* additionally containing α-tocotrienol. These attributes make them well suited for soaps, bio-lubricants, and antioxidant-enriched skincare formulations.

Aril oils, being richer in PUFAS and α-tocopherol, offer anti-inflammatory and barrier-repairing properties valuable for functional foods and therapeutic cosmetics. The high α-tocopherol (91.8 mg/kg) and tocotrienols in *T. dregeana* aril oil exceed many conventional plant oils, though stabilization is needed to address oxidative susceptibility.

Collectively, these findings support a dual-oil strategy: stable kernel oils for structural and long shelf-life products, and bioactive-rich aril oils for therapeutic and nutraceutical uses. Compared with other African oils, *Trichilia* stands out for its rare blend of oxidative stability, diverse fatty acid composition, and exceptional vitamin E density, positioning it as a valuable and underutilized African resource for high-value product development.

Importantly, when benchmarked against Codex Alimentarius guidelines, pharmacopoeial standards, and ISO cosmetic requirements, this study confirms that *Trichilia* oils—particularly kernel oils—already meet, or can readily be refined to meet, international thresholds for safety and stability. Kernel oils are thus immediately market-ready for cosmetic and industrial uses, while aril oils, despite requiring stabilization, provide distinctive PUFA-rich and vitamin E–dense profiles that align with nutraceutical and therapeutic niches. This standards-based validation strengthens the case for *Trichilia* oils as differentiated African resources with strong global commercial appeal.

## 6. Future Research Directions

While this study has clarified the yield, composition, and functional potential of *Trichilia emetica* and *T. dregeana* seed oils, several knowledge gaps remain that warrant future investigation. First, systematic evaluation of agronomic and ecological factors—such as soil type, rainfall, and seasonal variation—on oil yield and composition would help optimize cultivation and harvesting strategies. In particular, overcoming the limitation of generalization imposed by data collected in a single season requires multi-season and multi-location studies to capture temporal and spatial variability. Second, detailed characterization of bioactive compounds in the cake residues, including limonoids and saponins, is needed to support their valorization in animal feed, biofertilizers, and pharmaceutical applications. Third, controlled storage and formulation studies should be undertaken to address oxidative stability challenges, particularly in PUFA-rich aril oils, and to test antioxidant fortification strategies for maintaining product quality. Finally, comprehensive toxicological and clinical assessments are required to establish safety profiles for nutraceutical and cosmeceutical uses, thereby facilitating regulatory approval and market entry. Addressing these research directions will not only strengthen the scientific basis for *Trichilia* oil utilization but also accelerate its integration into sustainable value chains that benefit both local communities and industry.

## Figures and Tables

**Figure 1 plants-14-03071-f001:**
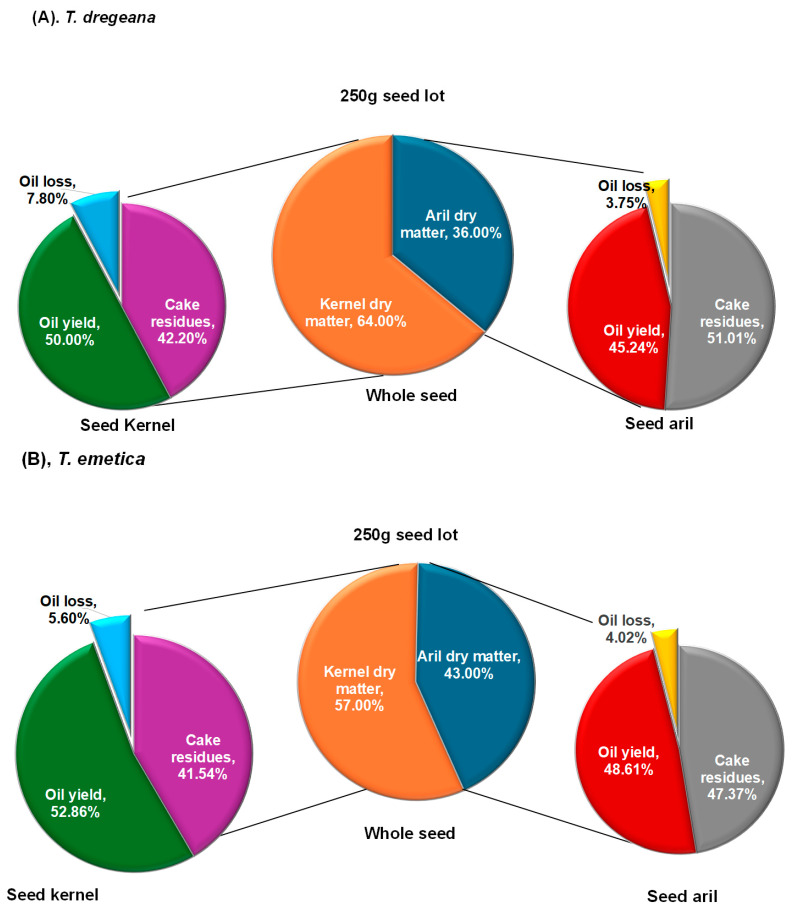
Partitioning of 250 g seed dry matter (%) between kernels and arils, oil and press residues, and associated oil loss during extraction in (**A**) *T. dregeana* and (**B**) *T. emetica*. HSD_0_._05_ values were 1.50 for dry matter, 7.64 for press residues, 8.34 for oil yield, and 2.65 for oil loss.

**Figure 2 plants-14-03071-f002:**
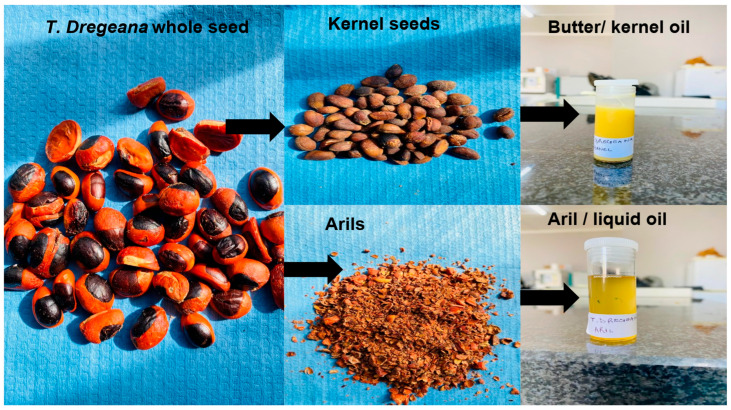
Mechanical separation and extraction of kernel and aril oils from *Trichilia dregeana* using Karaerler Makina’s NF80 screw press. The kernel-derived oil solidifies into a butter-like consistency, whereas the aril-derived oil remains liquid at room temperature.

**Table 1 plants-14-03071-t001:** Average number of hand-counted seeds per 200 g for *Trichilia emetica* and *Trichilia dregeana* (each replication was from a different tree).

Species	Number of Seeds Per 200 g					
	Rep 1	Rep 2	Rep 3	Rep 4	Rep 5	Average
*Trichilia emetica*	594	616	584	605	614	603
*Trichilia dregeana*	158	156	158	161	162	159

**Table 2 plants-14-03071-t002:** Physiochemical properties of *Trichilia emetica* and *Trichilia dregeana* kernel and aril oils.

Physiochemical Parameter	*T. dregeana* Aril Oil	*T. emetica* Aril Oil	*T. dregeana* Kernel Oil	*T. emetica* Kernel Oil
Peroxide Value (meq/kg oil)	12.0 ± 2.2	10.3 ± 1.9	LOD = 0.24	LOD = 0.24
Acid Value (mg KOH/g sample)	7.4 ± 1.2	12.0 ± 1.9	4.40 ± 0.68	4.45 ± 0.69
Free Fatty Acid (g oleic acid/100 g oil)	3.73 ± 0.58	6.01 ± 0.93	2.21 ± 0.34	2.24 ± 0.35
Anisidine value (mm/kg oil)	0.65 ± 0.11	2.70 ± 0.44	LOD = 0.16	LOD = 0.16
Unsaponifiable Matter (g/100 g)	Lower than LOQ	0.423 ± 0.08	1.00 ± 0.18	0.77 ± 0.14
Iodine Value	79.5 ± 1.4	72.0 ± 1.3	56.8 ± 1.0	45.00 ± 0.86
Saponification Value (mg KOH)	186.3 ± 3.4	186.0 ± 3.4	188.8 ± 3.4	197.3 ± 3.6
Relative Density (20 °C)	0.916	0.912	0.921	0.92
Refractive Index (20°)	1.467	1.465	1.465	1.46

Values represent the mean ± standard deviation of physicochemical parameters measured for oils extracted from the arils and kernels of *Trichilia dregeana* and *Trichilia emetica*. LOD—indicates the limit of detection, LOQ—Limit of quantification, FFA—Free Fatty Acids; KOH—Potassium Hydroxide; meq—milliequivalents; mg—milligrams; g—grams;; mm—milli-absorbance units; °C—degrees Celsius.

**Table 3 plants-14-03071-t003:** Fatty acid composition of *Trichilia dregeana* and *Trichilia emetica* seed aril and kernel oils extracted using screw a press.

Fatty Acid	*T. dregeana* Aril Oil	*T. emetica* Aril Oil	*T. dregeana* Kernel Oil	*T. emetica* Kernel Oil
C15:0	0.60	0.60	None detected	None detected
C16:0 P	28.7 ± 3.3	38.1 ± 4.4	40.4 ± 0.95	51.8 ± 6.0
C16:1 n7	Lower than LOQ	Lower than LOQ	0.550 ± 0.090	0.84 ± 0.14
C17:0	Lower than LOQ	Lower than LOQ	ND	Lower than LOQ
C18:0	3.17 ± 0.16	2.92 ± 0.15	2.25 ± 0.11	1.85 ± 0.09
C18:1 n9	40.5 ± 1.2	31.40 ± 0.94	47.6 ± 1.4	36.4 ± 1.1
C18:2 n6	24.31 ± 0.67	25.19 ± 0.69	7.89 ± 0.22	7.88 ± 0.22
C18:3 n3	0.550 ± 0.039	0.350 ± 0.025	0.560 ± 0.040	0.45 ± 0.03
C20:0	Lower than LOQ	Lower than LOQ	Lower than LOQ	Lower than LOQ
C20:1 n9	0.68 ± 0.13	0.370 ± 0.068	Lower than LOQ	Lower than LOQ
Unknown	0.60	ND	ND	-

Values represent the mean ± standard deviation of individual fatty acid components identified in aril and kernel oils of *Trichilia dregeana* and *Trichilia emetica*. Fatty acids are reported using shorthand notation for carbon chain length and degree of unsaturation, with corresponding common names provided. ND—none detected; LOQ—Limit of quantification; C15:0—Pentadecanoic acid; C16:0—Palmitic acid; C16:1 n7—Palmitoleic acid; C17:0—Margaric acid; C18:0—Stearic acid; C18:1 n9—Cis Oleic acid; C18:2 n6—Cis Linoleic acid; C18:3 n3—Linolenic acid; C20:0—Arachidic acid; C20:1 n9—Eicosenoic acid.

**Table 4 plants-14-03071-t004:** Tocols identified from *Trichilia dregeana* and *Trichia emetica* kernels and aril oils extracted using a screw press.

Tocol	*T. dregeana* Aril Oil	*T. emetica* Aril Oil	*T. dregeana* Kernel Oil	*T. emetica* Kernel Oil
α-tocopherol	91.8 ± 8.5	54.1 ± 5.0	52.6 ± 4.9	46.3
α-tocotrienol	Lower than LOQ	Lower than LOQ	79.5 ± 8.7	Lower than LOQ
β-tocopherol	Lower than LOQ	Lower than LOQ	Lower than LOQ	ND
γ-tocopherol	53.9 ± 5.9	41.0 ± 4.5	183 ± 20	202
γ-tocotrienol	15.8 ± 1.4	21.9 ± 1.9	68.1 ± 6.0	34.3
β-tocotrienol	Lower than LOQ	Lower than LOQ	24.1 ± 3.6	7.6
δ-tocopherol	Lower than LOQ	ND	Lower than LOQ	Lower than LOQ
δ-tocotrienol	Lower than LOQ	Lower than LOQ	73.27	32.60
Total	182 ± 18	130 ± 13	486 ± 48	335
Vitamin E activity (α-TE) *^,#^	97.20	58.15	94.71	66.52

Values represent the mean ± standard deviation of individual tocols (mg/kg oil) present in the aril and kernel oils of *T. dregeana* and *T. emetica*. * Not Sanas accredited. ^#^ Vitamin E activity was calculated based on α-tocopherol equivalents. ND—none detected; LOQ—Limit of Quantification; α-TE—Alpha-Tocopherol Equivalent; α—alpha; β—beta; γ—gamma; δ—delta.

**Table 6 plants-14-03071-t006:** Methods employed for the physicochemical analyses of *Trichillia emetica* and *Trichillia dregeana* aril and kernel oil.

Method Name	Method	References
Peroxide Value	AOCS Method CD 8-53	[45]
Free Fatty Acids/Acid Value	Based on AOCS Method Cd 3d-63	[45]
Saponification Value	AOCS Method Cd 3-25	[45]
Fatty Acid profile	Based on AOCS Ce 2-66	[45]
Iodine Value	AOCS Method Cd 1c-85	[45]
Unsaponifiable Material	AOCS Method Ca 6a-40	[45]
Anisidine Value	AOCS Method Cd 18-90	[45]
Tocopherols	ISO 9936	[46]
Relative density *	Guy Lussac Pyknometer	Not applicable
Refractive index *	Atago refractometer	Not applicable

* Not South African National Accreditation System (SANAS) accredited (all other methods are SANAS accredited). The methods used were conducted and recognized analytical protocol. All references are based on official AOCS methods. This incorporates the AOCS Method. CD 8-53 for peroxide, Cd 3d-63 for acid, Cd 3-25 for saponification, Ce 2-66 for fatty acid profiling, Cd 1c-85 for iodine, Ca 6a-40 for unsaponifiable material, and Cd 18-90 for anisidine value. Tocopherol content was determined using ISO 9936 [46]. Relative density and refractive index were determined using a Guy Lussac pyknometer and an Atago refractometer. AOCS—American Oil Chemists’ Society; ISO—International Organization for Standardization. Cd—Chemistry determination, Ce—Chromatography ester, Ca—Chemistry analytical and CD—Chemistry Determination.

## Data Availability

The research data can be requested from the authors.

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
