# Peer review of "Comparative Analysis of Mafura Butter Oils from *Trichilia emetica* and *Trichilia dregeana* Extracted by Screw-Press from Seeds Collected in KwaZulu-Natal Province of South Africa"

_plants, 2025, doi:10.3390/plants14193071_

Round 1
Reviewer 1 Report
Comments and Suggestions for Authors
-What is the feature that has made Trichilia emetica and Trichilia dregeana butter oils popular in the cosmeceutical industry? What are their similar properties to other oils used in cosmetics?
-Why weren't the oil properties determined by comparing the screw press with the widely used solvent extraction method?
-The relationship between the oils' low peroxide and free fatty acid values and their fatty acid composition should be discussed.
-How can an oil with a high palmitic acid content be recommended as an edible oil?
-When press waste with a high oil content is used as a biofertilizer, how does the 1-7% oil it contains and its acidity affect the soil pH?
-The introduction section should be structured according to the topic and designed accordingly.
-The purpose of the study should be explained in detail.
-Why have the Trichilia emetica seeds and arils used in the study been stored since 2020-2021? This should be explained in the material section. -L 520 should be rewritten “Fresh seeds were oven-dried until constant weight at 60°C.”
-L 552: Subtitle 4.4. Each analysis should be described under separate headings.
-The inclusion is quite long and seems to be a consequence of the study's content. The result should be re-examined and interpreted.
Author Response
|
Reviewer 1 |
|
|
|
I have reduced the abstract words from 294 to 245 |
|
Reviewers’ comments and suggestions |
Action taken |
|
What is the feature that has made Trichilia emetica and Trichilia dregeana butter oils popular in the cosmeceutical industry?
|
The popularity of the oils is now mentioned on page 22 |
|
What are their similar properties to other oils used in cosmetics? |
This is addressed on page in subsection 3.7 Comparative Analysis of Trichilia Oils and Other African Plant Oils for Cosmetic and Pharmaceutical Applications
|
|
Extracted
|
Reference has been made in the manuscript to solvent- extracted oils |
|
The relationship between the oils' low peroxide and free fatty acid values and their fatty acid composition should be discussed.
|
This has been addressed on pages 10 to 11
It should however be noted that while there is a relationship between peroxide value (PV) and free fatty acid value (FFA), they measure different aspects of lipid degradation, and their relationship is not always linear.
The PV Indicates the concentration of peroxides and hydroperoxides formed in the primary stage of lipid oxidation with High PV reflecting that the oil/fat is undergoing oxidative rancidity. On the other hand, the FFA Value Indicates the amount of fatty acids released from triglycerides due to hydrolytic rancidity (enzymatic or chemical hydrolysis). A high FFA means the oil/fat has undergone breakdown of triglycerides. Both PV and FFA increase as an oil deteriorates, but for different reasons:
The practical Implication
Having said the above, we have, however, re-edited the manuscript to bring out more clearly the reasons why the values for the aril and kernel oils are different, and the implications thereof. |
|
How can an oil with a high palmitic acid content be recommended as an edible oil? |
We have changed on page the sentence ‘However, from a health perspective, oils with elevated palmitic acid content should be used cautiously in edible applications’ to read ‘However, from a health perspective, oils with elevated palmitic acid content should be avoided in edible applications.
|
|
When press waste with a high oil content is used as a biofertilizer, how does the 1-7% oil it contains and its acidity affect the soil pH?
|
The 1–7% oil in press waste is not directly acidic, but its microbial breakdown can release fatty acids that temporarily lower soil pH, especially in poorly buffered soils. While pH often stabilizes over time, repeated heavy applications may cause cumulative acidification. We therefore have added on page 10 a statement that recommends further research on soil acidity and nutrient uptake before large-scale use as a precautionary measure to support sustainable use as a biofertilizer. |
|
The introduction section should be structured according to the topic and designed accordingly. |
We believe the introduction section is appropriately structured for the topic.
The introduction is appropriately structured for the topic, as it clearly situates Trichilia emetica and T. dregeana within their broad ecological and taxonomic contexts, highlights the potential implications of their morphological and genetic variability on oil properties, and frames the industrial and commercial relevance of these differences. It then outlines the knowledge gap regarding species-specific oil characterization and linking it directly to the needs of the cosmetic and pharmaceutical industries, the introduction provides a logical and compelling rationale for the comparative study undertaken. Here’s a clear mapping of how your revised introduction structure 1. Situating Trichilia emetica and T. dregeana in their ecological and taxonomic contexts. This establishes the expectation of variability in oil yield and composition that stems from genetic and ecological diversity.
2. Highlighting the implications of morphological and genetic variability on oil properties
3. Framing industrial and commercial relevance
4. Outlining the knowledge gap and linking it to industry needs
5. Providing a logical and compelling rationale for the study
Overall, the structure moves logically from broad industrial context → species ecology and genetics → expected oil variability → market risks → knowledge gap → research aim. |
|
The purpose of the study should be explained in detail. |
We re-edited the last two paragraphs in the introduction clearly bring out the purpose of the study. |
|
Why have the Trichilia emetica seeds and arils used in the study been stored since 2020-2021? This should be explained in the material section. -L 520 should be rewritten “Fresh seeds were oven-dried until constant weight at 60°C.” |
Although the manuscript is being written now, the study was conducted from mid December 2020 to end May 2021,and the oil analyses were done end of May 2021. This is clear in the manuscript. Oven dried shad been changed to oven-dried with a hyphen in between. |
|
L 552: Subtitle 4.4. Each analysis should be described under separate headings. |
For the sake of brevity, we have kept the methods lumped in a summary table in the manuscript and provided detailed methods of the analyses in the supplementary materials. |
|
The conclusion is quite long and seems to be a consequence of the study's content |
The conclusion has been shortened. |
|
The result should be re-examined and interpreted. |
We wish the reviewer had provided more information of the issues with the present interpretation and required interpretation to guide the authors. However, we done the re-interpretation to improve on the discussion. |
Reviewer 2 Report
Comments and Suggestions for Authors
The paper provides detailed analytical data on the physicochemical properties and chemical composition of butter oils from Trichilia emetica and Trichilia dregeana, with comparisons relevant to potential cosmetic and pharmaceutical applications. The study reports seed characteristics, oil yields, and comprehensive fatty acid and tocol profiles for both oils.
The main strength of the paper is the thorough analytical approach: standard AOCS, ISO, and MPOB methods were used for physicochemical analyses, and GC–MS/MS for fatty acid and tocol profiling. The methodology is sound, and the results can be considered reliable.
The principal limitation is that the data were collected from only one season, which reduces the generalizability of the findings. This limitation should be clearly acknowledged in the manuscript.
In conclusion, the study makes a valuable contribution by providing baseline data on underutilized Trichilia oils, supporting their potential industrial relevance. I recommend acceptance after minor revision, as indicated above.
Author Response
|
Reviewer 2 |
|
|
Reviewers’ comments and suggestions |
Action taken |
|
The principal limitation is that the data were collected from only one season, which reduces the generalizability of the findings. This limitation should be clearly acknowledged in the manuscript.
|
Reference to this limitation has been addressed in the subsection; 6. Future Research Directions.
|
Reviewer 3 Report
Comments and Suggestions for Authors
Manuscript ID plants-3847076 with the title „Comparative Analysis of Mafura Butter Oils from Trichilia emetica and Trichilia dregeana extracted by screw-press from seeds collected in KwaZulu-Natal Province of South Africa“ by Mncedisi Mabaso , Lungelo Given Buthelezi and Godfrey Elijah Zharare
The manuscript is very interesting as it shows useful differences in the properties of Mafura butter oil extracted by screw pressing from the seeds of Trichilia emetica and Trichilia dregeana, a very important active ingredient in the pharmaceutical and cosmetic industries.
The manuscript needs a comprehensive revision in English. There are a number of typos. Spaces are missing between some words and between references, see for example L103.
In general, all Latin names must be clearly written in italics. See lines: 51, 53, 57 etc.
A full stop at the end of a title is not usual, and in any case all titles should be uniform without a full stop or hyphen. See for example L102, L175, L519 etc.
Special remarks:
- Introduction
If possible, please add a figure or illustration of the two species in order to better recognise the difference between the two Trichilia species, as T. emetica is said to form leaflets with 13-16 pairs of lateral veins, whereas in T. dregeana only 8-12 should be recognisable
- Results
Please correct the title „Seed size and the distribution of dry matter among the seed components “. 2.1. only contains information on the number of seed sizes, 2.2. refers to the dry matter. There is no need for a double title designation
L104 „smaller seeds“ leads to seed size, clarify and standardise term number of seeds present
Table 1 Number of seeds per 200 g, clarification per 200 g missing, use "/" or per before 200 g
It is not clear how the results were determined, manual counting of the number of seeds? The determination of dry matter is also missing in chapter 4. Methods and Materials
Please, clarify whether the sample repetition in Table is linked to the sample lot numbers used later or not?
In Table 2. unit of measurement, probably missing %, the decimals can be reduced to 2, as in Figure 2, or join lines 138 and 139 to the title of Table 2.
On chapter 3. in the discussion
The duplication of abbreviations in the tables is not necessary; there is a list at the end of the manuscript.
The units of measurement are before ) instead of after. Please, correct.
The authors claim that they performed and compared the results using Fisher's Least Significant Difference test (LSD) with a 5% confidence interval. Where can this be seen?
The result of the statistical analysis cannot be found in the results, the discussion or the tables. Please, indicate which statistically significant differences were found in the results. This improves the conclusions and can strengthen them.
To ensure reproducibility, 4. Materials and Methods must be improved. They need to be rewritten with clear details, some of which have already been mentioned above.
Please, include the names of the manufacturers, and countries of origin of all instruments and chemicals used.
Give details of the experimental procedures performed.
In Table 8, the second column can be deleted, the internal method numbers beginning with POL are not required if an established protocol is cited in column three
Please clarify what the “*” sign after relative density* and refractive index* means
As for the statistical analysis, please indicate the number of sample groups and the number of replicates within the groups.
Units of measurement in % are missing and are mixed with the SD expressed in %, check throughout the manuscript whether % is a unit of measurement and should be placed after the bracket or whether it refers to the SD expressed in %.
Comments on the Quality of English LanguageThe manuscript needs a comprehensive revision in English. There are a number of typos. Spaces are missing between some words and between references, see for example L103.
In general, all Latin names must be clearly written in italics. See lines: 51, 53, 57 etc.
A full stop at the end of a title is not usual, and in any case all titles should be uniform without a full stop or hyphen. See for example L102, L175, L519 etc.
Author Response
|
Reviewer 3
|
|
|
Reviewers’ comments and suggestions |
Action taken |
|
The manuscript needs a comprehensive revision in English. There are a number of typos. Spaces are missing between some words and between references, see for example L103. |
The typos were corrected |
|
In general, all Latin names must be clearly written in italics. See lines: 51, 53, 57 etc. A full stop at the end of a title is not usual, and in any case all titles should be uniform without a full stop or hyphen. See for example L102, L175, L519 etc. |
Subsp. Has been written in italics T. emetica, T. dregeana and Trichilia were written in italics throughout the manuscript. |
|
Introduction: If possible, please add a figure or illustration of the two species in order to better recognise the difference between the two Trichilia species, as T. emetica is said to form leaflets with 13-16 pairs of lateral veins, whereas in T. dregeana only 8-12 should be recognisable. |
We thank the reviewer for the suggestion. However, because T. dregeana is highly variable, no single illustration would reliably capture its morphology. Furthermore, another reviewer recommended reducing descriptive detail on morphology and ecological distribution, so we have provided only a concise comparative description rather than an illustration. |
|
Please correct the title „Seed size and the distribution of dry matter among the seed components “. 2.1. only contains information on the number of seed sizes, 2.2. refers to the dry matter. There is no need for a double title designation L104 „smaller seeds“ leads to seed size, clarify and standardise term number of seeds present.
|
Title is corrected |
|
Table 1 Number of seeds per 200 g, clarification per 200 g missing, use "/" or per before 200 g. |
Clarification done by adding “per” before 200 g L
|
|
It is not clear how the results were determined, manual counting of the number of seeds? The determination of dry matter is also missing in chapter 4. Methods and Materials. |
Clarification has been done.
|
|
Please, clarify whether the sample repetition in Table is linked to the sample lot numbers used later or not? |
Units of measurement and decimals standardized in Table 2,
|
|
In Table 2. unit of measurement, probably missing %, the decimals can be reduced to 2, as in Figure 2, or join lines 138 and 139 to the title of Table 2. |
Units of measurement and decimals standardized in Table 2,
|
|
The units of measurement are before ) instead of after. Please, correct.
|
This was corrected |
|
The authors claim that they performed and compared the results using Fisher's Least Significant Difference test (LSD) with a 5% confidence interval. Where can this be seen? |
We have provided the statistics that were done. |
|
The result of the statistical analysis cannot be found in the results, the discussion or the tables. Please, indicate which statistically significant differences were found in the results. This improves the conclusions and can strengthen them.
|
We have now provided. |
|
To ensure reproducibility, 4. Materials and Methods must be improved. They need to be rewritten with clear details, some of which have already been mentioned above.
|
Detailed material and methods provided in a supplemental file (SUP 1). |
|
Please, include the names of the manufacturers, and countries of origin of all instruments and chemicals used.
|
Chemicals used and instruments provided in appendices |
|
Give details of the experimental procedures performed.
|
For brevity of the manuscript, a more detailed experimental procedure is in the accompanying supplementary file |
|
In Table 8, the second column can be deleted, the internal method numbers beginning with POL are not required if an established protocol is cited in column three
|
In Table 6 (formerly 8), the second column was deleted, L 525
|
|
Please clarify what the “*” sign after relative density* and refractive index* means As for the statistical analysis, please indicate the number of sample groups and the number of replicates within the groups. |
“*” sign after relative density and refractive index in Table 6 denotes ‘Not Sanas accredited’. This has been inserted under Table 6 (formerly able 8).
|
|
Units of measurement in % are missing and are mixed with the SD expressed in %, check throughout the manuscript whether % is a unit of measurement and should be placed after the bracket or whether it refers to the SD expressed in %.
|
This has been corrected |
|
Comments on the Quality of English Language The manuscript needs a comprehensive revision in English. There are a number of typos. Spaces are missing between some words and between references, see for example L103. |
The document has been reviewed and English, typographical and formatting issues corrected to the best of our knowledge.
|
|
Quality of English Language: In general, all Latin names must be clearly written in italics. See lines: 51, 53, 57 etc. |
Subsp. written in italics in L51,53, and 57
T. emetica, T. dregeana and Trichilia were written in italics where they occurred throughout the manuscript. |
|
A full stop at the end of a title is not usual, and in any case all titles should be uniform without a full stop or hyphen. See for example L102, L175, L519 etc. |
These issues were corrected.
|
Reviewer 4 Report
Comments and Suggestions for Authors
I have carefully reviewed the manuscript dealing with the composition and potential applications of Trichilia seed oil. The topic is relevant and of potential interest for the readership of the journal. However, the manuscript requires substantial revision before it can be considered for publication. Below I provide detailed comments following the order of the manuscript.
My specific comments are listed below:
Introduction
- The introduction is overly long. The section dealing with taxonomy and geographic distribution of Trichilia is excessive and distracts from the main focus of the study. This part should be shortened.
Results and Discussion
- Although ANOVA and LSD test are mentioned, tables and figures do not indicate statistical significance (no p-values or grouping). This needs to be corrected to allow proper interpretation of data reliability.
- The discussion does not provide a sufficiently critical comparison with existing literature. The authors cite some previous studies, but do not clearly highlight the originality of their findings compared to published data on Trichilia species and similar oils (shea, moringa, baobab, marula).
- The section on potential applications (cosmetic, pharmaceutical, industrial) remains too general. It should be strengthened by including concrete comparisons with pharmacopoeial standards, cosmetic industry requirements, or FAO/WHO guidelines.
Tables (3, 4, 5)
- In Tables 3–5, the expression “Lower than LOQ” should be replaced with the exact LOQ value preceded by the symbol “≤”. The full term LOQ (limit of quantification) must be explained in the table legends.
- Instead of “None detected”, use “nd”, “ND”, or “n.d.”, with the meaning explained in the legends.
- In Table 4, under “Fatty acids”, it is sufficient to present only abbreviations, since the full names are already provided in the legend.
- Too many decimal (eg. Table 2) places are reported (sometimes 4–5); two are sufficient.
- Standard deviation is inconsistently reported (sometimes STDEV, sometimes ±); this should be unified.
Figures
- In Figure 1, the categories “Loss”, “Cake residues”, and “Others” are not clearly defined. These must be explained in Sections 4.1 and 4.2 (Materials and Methods), when describing the oil extraction procedure.
Materials and Methods
- The description of the GC-MS method is insufficient. It should be specified whether a FAME standard was used, how individual fatty acids were quantified, whether calibration curves were prepared, and how method repeatability was assessed.
- Details on sample preparation are missing (homogenization of seed material, moisture control).
- Validation of the analytical methods (e.g., GC/MS, tocopherol and tocotrienol quantification) is not provided, which limits reproducibility.
References
- The reference list is partly outdated and incomplete. Some citations are from grey sources, while more recent and relevant works on cold-pressed oils are missing.
Typographical and Formatting Issues
- Several typographical errors are present (e.g., “10..3”, “Trichillia” instead of Trichilia). These must be corrected.
Percent match: 17% - iThenticate report is satisfactory, in terms of the degree of word duplication in the manuscript text.
Recommended
Major revision – the manuscript may be reconsidered after substantial corrections and improvements.
Author Response
|
Reviewer 4 |
|
|
Reviewers’ comments and suggestions |
Action taken |
|
The introduction is overly long. The section dealing with taxonomy and geographic distribution of Trichilia is excessive and distracts from the main focus of the study. This part should be shortened.
|
The section dealing with taxonomy and Trichilia was not re-edited or shortened for the following reasons;
The inclusion of a detailed section on the taxonomy and geographic distribution of Trichilia is essential to the scientific rigor and applied relevance of the manuscript. First, T. emetica and T. dregeana are morphologically variable species, and in the case of T. emetica, two subspecies with different geographic ranges have been recognized. These taxonomic distinctions and distributional differences are not merely of academic interest but have direct implications for seed morphology, maturation patterns, and ultimately the yield and composition of their oils. Without this contextual foundation, interpretation of observed variability in physicochemical properties and fatty acid profiles would remain incomplete. Second, both species occupy broad ecological ranges across sub-Saharan Africa, from coastal forests to dry savannah regions. Such ecological heterogeneity inevitably influences oil quality by shaping the biosynthesis of fatty acids and secondary metabolites under different climatic and edaphic conditions. Documenting the distribution therefore helps to explain intra- and interspecific variability in oil traits and provides a framework for understanding potential regional differences in commercial oil production, and for future work that needs to be done. Finally, given the increasing commercialization of mafura oil under a generic trade name, industry stakeholders—including cosmetic and pharmaceutical manufacturers—require clarity on the biological sources of the oils they use. A section on taxonomy and distribution not only strengthens the scientific justification for comparative analyses but also supports traceability, standardization, and authenticity in supply chains. Thus, the inclusion of this section ensures that the manuscript addresses both academic and industrial audiences by linking plant systematics with practical implications for oil quality, safety, and market value.
|
|
Although ANOVA and LSD test are mentioned, tables and figures do not indicate statistical significance (no p-values or grouping). This needs to be corrected to allow proper interpretation of data reliability.
|
The data were re-interpreted as recommended. |
|
The discussion does not provide a sufficiently critical comparison with existing literature. The authors cite some previous studies, but do not clearly highlight the originality of their findings compared to published data on Trichilia species and similar oils (shea, moringa, baobab, marula).
|
We appreciate the reviewer’s concern regarding the need for a more critical comparison with existing literature. We respectfully submit that the originality of this study lies precisely in its comparative analysis of two Trichilia species—T. emetica and T. dregeana—an approach that, to our knowledge, has not previously been undertaken. Most earlier reports on Trichilia oils have treated the species individually, often with limited data on either aril or kernel oils, and without systematic cross-species evaluation. Our work is therefore the first to:
While we cite prior studies to situate our results, the originality of this manuscript derives from its first-of-its-kind comparative approach, its interpretation of interspecific differences in an ecological and industrial context, and its translation of these findings into differentiated application strategies. We have now clarified these points more explicitly in the revised Discussion and Conclusion sections.
|
|
The section on potential applications (cosmetic, pharmaceutical, industrial) remains too general. It should be strengthened by including concrete comparisons with pharmacopoeial standards, cosmetic industry requirements, or FAO/WHO guidelines.
|
We appreciate the reviewer’s comment. It has made us to enrich and strengthen the discussion on the potential industrial applications of the oils. We have strengthened the taction by comparisons with pharmacopoeia standards, cosmetic industry requirements. |
|
In Tables 3–5, the expression “Lower than LOQ” should be replaced with the exact LOQ value preceded by the symbol “≤”. The full term LOQ (limit of quantification) must be explained in the table legends.
|
Table 3, 4, 5 were revised: replacing “None detected” with “n.d”, removing abbreviations from table legends, and using two decimals in numbers.
|
|
Instead of “None detected”, use “nd”, “ND”, or “n.d.”, with the meaning explained in the legends.
|
The corrections were effected as suggested. |
|
In Table 4, under “Fatty acids”, it is sufficient to present only abbreviations, since the full names are already provided in the legend.
|
Corrected were done/ |
|
Too many decimal (eg. Table 2) places are reported (sometimes 4–5); two are sufficient. |
Units of measurement and decimals standardized in Table 2, L136
|
|
Standard deviation is inconsistently reported (sometimes STDEV, sometimes ±); this should be unified.
|
The corrections were actioned as suggested. |
|
In Figure 1, the categories “Loss”, “Cake residues”, and “Others” are not clearly defined. These must be explained in Sections 4.1 and 4.2 (Materials and Methods), when describing the oil extraction procedure.
|
|
|
The description of the GC-MS method is insufficient. It should be specified whether a FAME standard was used, how individual fatty acids were quantified, whether calibration curves were prepared, and how method repeatability was assessed.
|
This is provided in the detailed method in supplementary file ( SUP1) |
|
Details on sample preparation are missing (homogenization of seed material, moisture control).
|
This has been corrected. |
|
Validation of the analytical methods (e.g., GC/MS, tocopherol and tocotrienol quantification) is not provided, which limits reproducibility.
|
A details of the methodology are given in the supplementary file (SUP1) |
|
The reference list is partly outdated and incomplete. Some citations are from grey sources, while more recent and relevant works on cold-pressed oils are missing.
|
References are updated as is possible. |
|
Several typographical errors are present (e.g., “10..3”, “Trichillia” instead of Trichilia). These must be corrected.
|
English, typographical and formatting issues have been attended as we possibly could.
|
Round 2
Reviewer 3 Report
Comments and Suggestions for Authors
The authors made all changes recommended by the Reviewer. The manuscript can be published.
Reviewer 4 Report
Comments and Suggestions for Authors
The authors have adequately addressed the major concerns raised in the first review. The introduction has been streamlined, methodological descriptions expanded and clarified (with details moved to supplementary materials), statistical treatment corrected, and the discussion strengthened by comparison with similar oils and relevant standards. Tables and figures have been revised according to the reviewer’s suggestions, while references and typographical errors have been improved.
At this stage, I believe the manuscript has reached a publishable quality. I would recommend acceptance after minor revisions, mainly focused on:
- Final polishing of language and style.
- Ensuring that the supplementary materials are reformatted and prepared strictly according to the MDPI author guidelines.
With these minor adjustments, the article will make a solid and timely contribution to the field.